# Experimental Investigation on Pouring Aggregate to Plug Horizontal Tunnel with Flow Water

**Gailing Zhang [1], Shuang Hui [2], Weixin Li [3] and Wanghua Sui [1],***

[1]  School of Resources and Geosciences, Institute of Mine Water Hazards Prevention and Controlling Technology, China University of Mining and Technology, Xuzhou 221116, China; gailing-zhang@cumt.edu.cn
[2]  Environmental Engineering Company, China Design Group Co., Ltd., Nanjing 210014, China; binghuyise@126.com
[3]  Geotechnical Engineering Survey and Design Institute, North China Engineering Investigation Institute Co., Ltd., Shijiazhuang 050021, China; huishuang07@126.com
*  Correspondence: suiwanghua@cumt.edu.cn; Tel.: +86-139-5219-9519

**Abstract:** This paper presents an experimental investigation on the main factors that influence the effects of pouring aggregate to plug a tunnel that has been inundated by groundwater to reduce the flow velocity. Moreover, a criterion for plugging the tunnel under infiltrating water to resist flow is proposed. A range analysis and analysis of variance both show that the influencing factors on the efficiency of plugging in descending order is the aggregate particle size, followed by initial velocity of the water flow, and then the water–solid mass ratio. The sedimentation process of the aggregate is likened to the deposition of solid particles into slurry in which the particles settle under gravitational force, thus accumulating at the bottom of the tunnel model due to the forces of the water flow and gravity. The critical velocity of the water that will transport the aggregate without settling can be used as a criterion to determine whether there has been a successful plug of the resistance to flow in the tunnel. The experimental results show that the critical velocity of fine aggregate is less than that of coarse aggregate, and the section with smaller sized aggregate or fine aggregate that resists water flow is flatter. In addition, the required minimum space between two pouring boreholes for a successful resistance to flow is discussed.

**Keywords:** groundwater inrush; aggregate; tunnel; critical velocity; plug

---

## 1. Introduction

Mine groundwater disasters have already caused a miserable life price and serious economic loss in the world. The flooding of tunnels in mines and underground spaces due to groundwater inrush is one of the most detrimental disasters, so that there is much urgency to quickly control the water infiltration and restore the operations of underground engineering or mining production after the incident. Grouting has been widely used to address this problem, but, before that is done, pouring an aggregate plug is a major prerequisite to effective grouting so as to plug the tunnel and stop the water flow, which even plays a key role sometimes when the tunnel has a large cross-section area (Table 1). The main purpose of pouring aggregate is to build a section that effectively resists downstream flow to stop the flow and convert it from a confined pipe flow to a seepage flow through a porous medium (aggregates) so that subsequent grouting can be implemented to seal off the tunnel completely. The preferred location of the plug is on a horizontal tunnel and an uphill where water flows upward. A section that successfully resists flow is formed because of the accumulation of aggregate in the tunnel, which is somewhat similar to the deposition of solid particles of slurry for pipeline

transport. Therefore, the accumulation of aggregate in a circular cross-section tunnel with flow water can be investigated by referring to hydraulic pipeline transport.

**Table 1.** Some case histories of plugging groundwater inrushes in tunnels by pouring aggregate in Chinese coalmines.

| Coal Mine | Time | Groundwater Resource and Pathway | Treatment Measure and Material Consumption | Sealing Effect |
|---|---|---|---|---|
| Longmen coalmine, Henan | Groundwater inrush in 11 December 1994; Pouring aggregates from 20 June 1994 to 22 August, then grouting until 9 September. | Karst aquifer of the Cambrian limestone, through a pathway of geologic structure, with a pressure of 3 MPa and a flowrate of 2200 m³/h. | Pouring aggregates to control the speed of groundwater flow, then grouting to seal the remained flow; 3100.5 m³ of aggregates and 757 tons of cement. | 97.1% |
| Renlou coalmine, Anhui | Groundwater inrush in 4 March 1996; Pouring aggregates from 25 April to 25 May, then grouting until finishing. | Karst collapsed column of the Ordovician limestone, with a pressure of 5 MPa and a flowrate of 11,854–34,570 m³/h. | Pouring aggregates to plug the flow in tunnel first, and then sealing off the collapsed column; 129.88 m³ of aggregates, 15,032 tons of cement. | 85–90%, forming a plugging seal with a length of 60 m at a depth of 420–480 m in the column. |
| Wucun coalmine, Henan | Groundwater inrush in 15 November 1999; Pouring aggregates from 18 January 2000 to 10 March, then grouting until 11 April. | Aquifer of the Ordovician limestone intersected by faults and Karst collapsed column, with a steady flowrate of 2145 m³/h, a maximum of 2378 m³/h. | Pouring aggregates to plug the flow in tunnel first, and then sealing off the collapse column; 1535 m³ of aggregates, 3182.6 tons of cement. | 97–100%. |
| Dongpang coalmine, Hebei | Groundwater inrush in 12 April 2003; Pouring aggregates from 10 May to 11 June, then grouting until 23 September. | Karst collapsed column of the Ordovician limestone, with a pressure of 5 MPa and an average flowrate of 7000 m³/h. | Pouring aggregates to plug the flow in tunnel first, and then sealing off the collapse column; 42,837 m³ of aggregates, 26,396 tons of cement. | 98.71%, forming a plugging seal with a length of 105 m. |
| Sanshuping coalmine, Shaanxi | Groundwater inrush in 7 August 2003; Pouring aggregates from 15 October to 9 November, then grouting until 9 April 2012. | Karst aquifer of the Ordovician limestone, with a pressure of 3 MPa and an average flowrate of 8000 m³/, a maximum of 13,200 m³/h. | Pouring aggregates to plug the dynamic flow in tunnel, 25,716 m³ of aggregates, mainly fine sand, 60,383 tons of cement. | 98.71%, forming a plugging seal with a width of 23.5 m and a height of 3 m in the tunnel. |
| Panji coalmine No. 2, Anhui | Groundwater inrush in 25 May 2017; Pouring aggregates from 20 June to 27 July, then grouting until 16 August. | Karst collapsed column of the Ordovician limestone, with a flowrate of 3024 m³/h. | Pouring aggregates to plug the dynamic flow in tunnel, 21,141 m³ of aggregates, mainly fine sand, 15,349 tons of cement and fly ash. | 100%, forming a plugging seal with a length of 34 m. |

Slurry pipeline transport is when solid materials are conveyed through closed pipelines with liquid (usually water) as the carrier. The transport of solid materials through pipelines has had nearly a hundred years of history since the successful pipeline transport of coal slurry in the USA. Studies on the movement of the solid–liquid two-phase flow in pipelines began with experiments. The state of the solid–liquid mixtures was categorized as three kinds of states, namely homogeneous, intermediate, and heterogeneous types of flow, based on the suspension flow state of the sand and gravel in the

water and the size of the particles [1]. The flow regime of solid materials in pipelines was categorized by Wasp et al. into two types of flow: homogeneous and heterogeneous [2]. Solid particles are labeled as bed load or suspended load based on the support force and the motion of the particles in solids [3]. The heterogeneous to homogeneous transition of slurry flow in pipes was studied by Miedema [4]. The transport of sand/water slurries along a horizontal pipeline has been the subject of many studies, for example those by Soepyan et al. [5] and Zouaoui et al. [6]. The pattern of the two-phase flow is related to many factors, but, in this paper, the classification of the pattern of flow proposed by Fei [3] is adopted.

The critical velocity is the minimum velocity of the water that will transition the aggregate particles from a static to dynamic state in order to determine the flow state of the aggregate particles in the pipe. Durand used the concept of "limit deposit velocity" to denote the critical deposit velocity, which was determined by the presence of siltation in the pipelines, and provided a representative Durand formula for calculation [1]. Graf et al. proposed the term "critical flow rate", which is defined as the velocity at which solid particles precipitate from a suspended state and form a fixed bed [7]. Fei proposed three types of critical velocities to classify the pattern of flow, including the critical velocity of the water that will transport the aggregate without settling, which is the velocity that the solid particles start to slide or roll on the bed surface [3]. A predictive model for the deposition velocity of slurry composed of fine particles was developed by Wasp and Slatter [8], in accordance with the assumptions made by Thomas [9], which had an important effect on obtaining the velocity formula. Pinto et al. offered a semi-empirical formula to predict the critical deposit velocity and analyzed the effect of the particle shape on the velocity through a sphericity function [10]. Bratland calculated the minimum velocity of slurry in inclined pipes [11]. Kim et al. studied the effect of the pipeline shape on the flow velocity of the deposition and found that the deposition-limit velocity in a square duct is smaller than that in a circular pipe [12]. The critical velocity required to initiate fine particle movement with water is less than that for coarse particle movement [13]. The term "critical velocity" is adopted in this paper to determine the limit velocity of solid particles that move from the static to dynamic state.

In the two-phase flow of the pipeline, the mechanism behind the solid particle concentration should be understood in order to illustrate friction loss and other issues [14]. The solid particle concentration and loss of resistance are correlated. Vlasák et al. evaluated the effect of the slurry velocity and solid particle concentration on flow behavior and reduction of pressure in the slurry in a turbulent flow state [15]. The experiments were conducted with natural, plastic, and emery sand to study the influence of different specific gravities on resistance with an empirical formula proposed by Durand [1]. Newitt et al. introduced a formula for calculating the hydraulic gradient of settling slurry based on studying the power and energy consumed by the suspension of solid particles [16]. Dimensional analysis methods were used to carry out the regression analysis on multitudinous data, and a computation model of the resistance loss in the pipe due to the flow of slurry was proposed to predict drops in pressure in the flow of solid–liquid suspensions in pipelines [17,18]. Miedema developed a head loss model for slurry transport in a heterogeneous regime based on energy considerations [19]. The experimental investigation of solids transport by Allahvirdizadeh et al. showed that increasing the fluid viscosity may not always be an effective means for addressing the problem of transport at high flow rates [20]. Duckworth and Argyros experimented with materials that are lighter and heavier than water to study their effects on the hydraulic gradient of particle density in the pipeline. Therefore, particle size and specific gravity are important variables for friction loss analyses [21].

Computational Fluid Dynamics has been widely used as an engineering-effective tool for slurry pipe flow design and management in recent years. The Eulerian–Lagrangian large eddy simulation method was used to investigate the dynamics of single-dispersed fine particle pipe slurry flow with a high concentration and turbulent liquid–solid slurries in horizontal pipes [22–26]. A new two-fluid model (TFM) for the numerical simulation of the flow of the whole suspended slurry in the horizontal tube was proposed by Messa et al. [27] and improved later by Messa and Malavasi [28].

Results of TFM were compared with the experimental data and further confidence was provided for the use of the TFM as an effective tool for engineering design [29].

The application of grouting to plug tunnels that have been inundated with water is generally a difficult task due to the obscurity and mainly relies on experience and adjustments made on the spot. Therefore, theoretical research on grouting and sealing technologies of tunnels with water inrush falls behind practical applicability. Consequently, the main purpose of this study was to experimentally investigate the deposition of aggregate into a modeled tunnel under the influence of different factors, including the initial velocity of the water flow, which is defined as the steady flow rate before pouring aggregate in the tunnel due to groundwater inrush; aggregate particle size; and water–solid mass ratio. Meanwhile, aggregate accumulation was also investigated on the basis of both experiment and slurry transport theory. The results will be helpful for a better understanding on the formation of aggregate siltation and effects of pouring aggregate for plugging, and, as such, a criterion for sealing the tunnel under infiltrating water to provide resistance to the water flow is proposed. To simplify the engineering geological conditions, aggregate was poured into a transparent pipe to model plugging the water flow in a tunnel.

## 2. Materials and Methods

### 2.1. Materials

The experimental materials consisted of aggregate with four different particle sizes: less than 0.1 mm (*B*1), 0.1–0.5 mm (*B*2), 0.5–2 mm (*B*3), and 2–5 mm (*B*4). Table 2 lists the particle size distribution of the aggregates used in the experiments. During the experiments, aggregates with different water–solid mass ratios were poured through a vertical borehole into the tunnel with no additional pressure except for gravity. The deposition of the aggregates provided the resistance to the water flow in the tunnel prototype.

### 2.2. Experimental Set-Up

The experiments were designed by using the Froude similarity criteria. The length scale ratio $\lambda_L$ between the prototype and the model was 20. The flow velocity scale ratio $\lambda_L^{\frac{1}{2}}$ between the prototype and the model was 4.47, in accordance with the Froude similarity criteria [30,31].

A transparent acrylic pipe with a pouring and monitoring system was used as the model in the orthogonal experiments. Figure 1 shows the photo and schematic diagram of the experimental set-up, which is composed of a tunnel prototype (labeled as 6), aggregate pouring system (9), data collection system (2, 4, 5, and 7), and water source (1, 3, and 8).

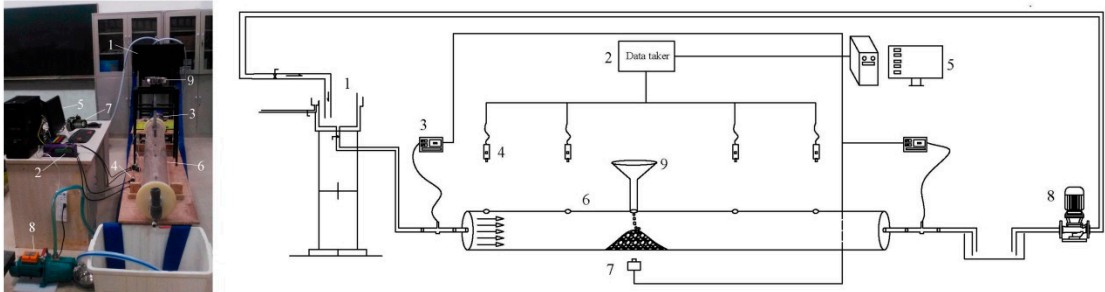

**Figure 1.** Photo (**left**) and schematic diagram (**right**) of experimental set-up. (1) water supply device; (2) data collection instrument; (3) water meter; (4) pressure sensor; (5) computer; (6) tunnel replica; (7) camera; (8) water pump; and (9) funnel for pouring aggregate.

The transparent acrylic tunnel replica has a length of 2000 mm, an inner diameter of 190 mm, and wall thickness of 5 mm. A borehole used for pouring with a diameter of 25 mm was drilled on the tunnel roof 800 mm in the horizontal direction from the entrance of the water flow. Four boreholes with a diameter of 8 mm, which were used to detect water pressure, were drilled at a horizontal distance of 400, 600, 1100, and 1400 mm from the entrance of the water flow. Four water pressure sensors were connected to the boreholes with polyurethane tubing. The tunnel replica was fixed onto a support made of wood material. Figure 2 is the schematic diagram of the tunnel prototype.

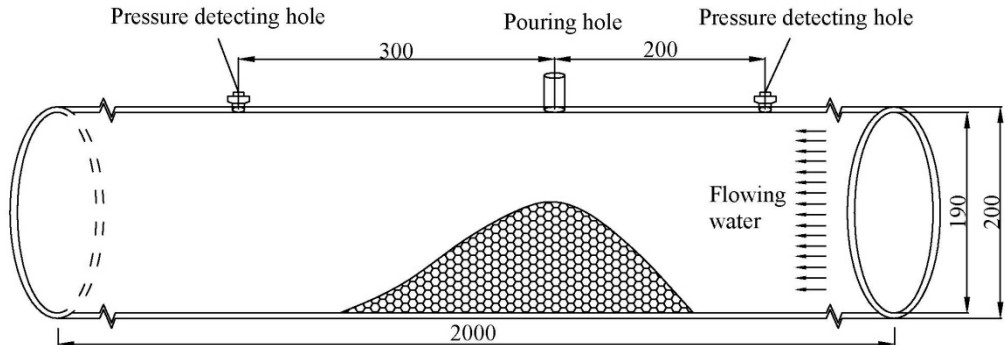

**Figure 2.** Schematic diagram of tunnel prototype with water flow (unit: mm).

A water storage tank with a constant water head was placed in the inlet, and the water head difference between the storage tank and the tunnel was 0.7 m. Water flowed from the storage tank into the tunnel through a valve. A flow meter was used to measure water flux and then the velocity of water flow was defined as the flow rate in the cross-sectional area of the residual channel in a unit time. The flow rate could be regulated from 0 to 2.0 m³/h by controlling the valve, and then the initial flow velocity of the water could be subsequently adjusted. The water supply system simulated the dynamic modeling of water inrush with a constant head and controlled flow rate.

The saturated aggregate was poured into the tunnel with different water–solid mass ratios at a flow rate of approximately 20 g/s until the maximum height of the aggregate was reached. The pouring of the aggregate and its deposition and movement were captured with a camera placed on the side of the tunnel.

### 2.3. Experimental Design

The efficiency of plugging the tunnel with aggregate to resist water flow are influenced by many factors, such as the shape of the tunnel, aggregate material, velocity of the water flow, water–solid mass ratio, and distance between the grouting boreholes. In addition, after the completion of pouring the aggregate, the timing and means of the grouting are still important factors. Several influential factors that affect the plugging during pouring aggregate were chosen for examination in this study, including the initial velocity of the water flow (denoted as *A* in the test number), aggregate particle size (*B*), and water–solid mass ratio (*C*).

Orthogonal arrays were adopted in the design, which is an experimental method for investigating the influence that different factors do on the estimated indexes under different levels. The orthogonal arrays had a total of 16 experiments with three factors, each changing at four levels (denoted as 1–4) in the study. For example, *A*1*B*1*C*1 means that *A*, *B*, and *C* were at the first level with the specific values listed in Table 2.

Table 2. Orthogonal experimental matrix on modeling of pouring aggregate into tunnel.

| Trial No. | Symbol for Trial | Initial Velocity of Water Flow (cm/s) *A* | Particle Size of Aggregate (mm) *B* | Water–Solid Mass Ratio *C* | Cross-Section Area of Residual Water Channel (mm$^2$) | Efficiency of Plugging (%) *PE* |
|---|---|---|---|---|---|---|
| 1 | *A1B1C1* | 0 | <0.1 | 1 | 271.05 | 99.04 |
| 2 | *A1B2C2* | 0 | 0.1–0.5 | 1.5 | 332.44 | 98.83 |
| 3 | *A1B3C3* | 0 | 0.5–2.0 | 2 | 491.11 | 98.27 |
| 4 | *A1B4C4* | 0 | 2.0–5.0 | 3 | 592.13 | 97.91 |
| 5 | *A2B1C2* | 0.5 | <0.1 | 1.5 | 553.32 | 98.05 |
| 6 | *A2B2C1* | 0.5 | 0.1–0.5 | 1 | 577.92 | 97.96 |
| 7 | *A2B3C4* | 0.5 | 0.5–2.0 | 3 | 770.98 | 97.28 |
| 8 | *A2B4C3* | 0.5 | 2.0–5.0 | 2 | 695.35 | 97.55 |
| 9 | *A3B1C3* | 1.0 | <0.1 | 2 | 379.39 | 98.66 |
| 10 | *A3B2C4* | 1.0 | 0.1–0.5 | 3 | 634.79 | 97.76 |
| 11 | *A3B3C1* | 1.0 | 0.5–2.0 | 1 | 1042.93 | 96.32 |
| 12 | *A3B4C2* | 1.0 | 2.0–5.0 | 1.5 | 1034.04 | 96.35 |
| 13 | *A4B1C4* | 1.5 | <0.1 | 3 | 371.72 | 98.69 |
| 14 | *A4B2C3* | 1.5 | 0.1–0.5 | 2 | 577.35 | 97.96 |
| 15 | *A4B3C2* | 1.5 | 0.5–2.0 | 1.5 | 851.84 | 97.00 |
| 16 | *A4B4C1* | 1.5 | 2.0–5.0 | 1 | 991.38 | 96.50 |

The flow velocity has an important influence on the plugging of the tunnel with water flow. The initial flow velocity was chosen as 0–1.5 cm/s, which is equal to an actual water flow velocity from 0 to 6.7 cm/s in prototype. This simulates steady flow with a slowing velocity in practice. A higher flow velocity means a greater water carrying capacity. Each aggregate particle is therefore subject to a critical velocity of the water that will transport it without settling. Therefore, the aggregate particle size is correlated to the flow velocity [32]. Aggregate with a particle size less than 5 mm was used in the investigation. The water–solid mass ratio was selected mainly based on the aggregate particle size and the experiment itself in order to pour the aggregate smoothly without blocking the pouring borehole. Therefore, the water–solid mass ratios were 1, 1.5, 2, and 3 in accordance with the experimental materials and pouring method, which are less than the water–solid mass ratios in real life practices that range from 5 to 10 [33].

## 3. Results

### 3.1. Shape of Deposited Aggregate

The aggregates were poured into the tunnel replica and gradually accumulated underneath the pouring borehole and moved along the tunnel. Pouring aggregate is a process in which the particles of the slurry settle and form a cone shape after the water–sand mixture is poured into the water in the tunnel. During the process of slurry settling, the aggregate particles are subjected to gravity, buoyancy, impulse force exerted by water, viscous drag forces, and other forces, which cause the aggregate deposits to move against the downstream direction of the water flow relative to the pouring hole of the aggregate. The offset distance of the deposited aggregate relative to the pouring hole is different depending on the aggregate particle size and the initial velocity of the water flow. Figure 3 shows the final shape of the deposition relative to the four different aggregate particle sizes in the tunnel. When viewed from the side, the deposition has a cone shape. Figure 3 shows that a smaller particle size results in a greater offset distance at the surface relative to the pouring hole. Therefore, the deposition shape is flatter. The shape of the deposited aggregate against the downstream side of the water flow is flatter than that against the upstream face of the water flow.

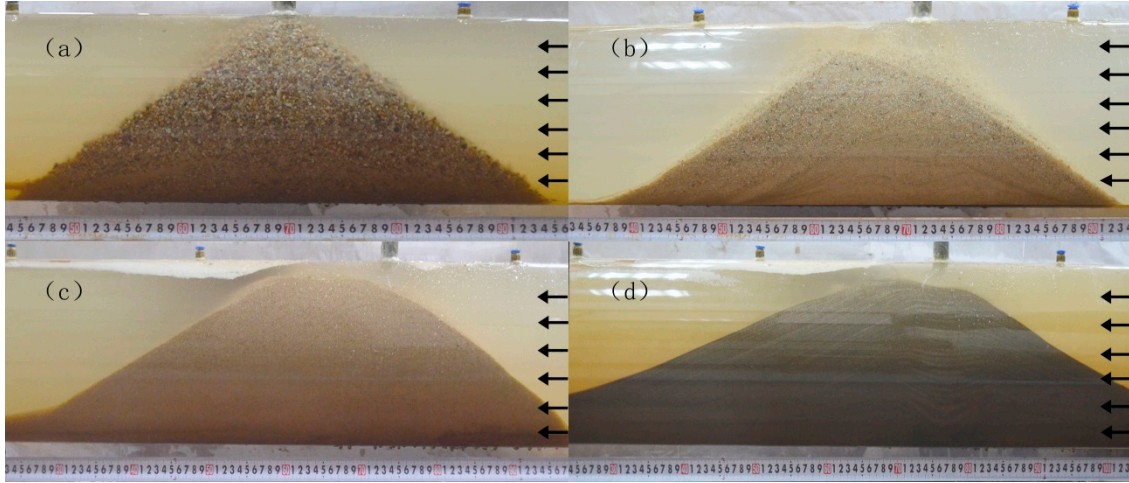

**Figure 3.** Deposition of aggregate with four different particle sizes (arrows represent water flow). Deposition of aggregate with particle size of: (**a**) 2–5 mm (Trial No. 4, *v* = 0); (**b**) 0.5–2 mm (Trial No. 11, *v* = 1.0 cm/s); (**c**) 0.1–0.5 mm (Trial No. 10, *v* = 1.0 cm/s); and (**d**) less than 0.1 mm (Trial No. 9, *v* = 1.0 cm/s).

### 3.2. Formation Process of Deposition with Cone Shape

Natural granular soils have an underwater angle of repose, which is similar to the angle of internal friction of the material [34]. The angle of the cone shaped deposit is also shown in Figure 4. The underwater angle of repose of the deposited aggregate against the downstream side is slightly less than that against the upstream under the force of the water flow. The shape of the deposited aggregate can be defined by the angle of repose in accordance with the maximum height of the deposited aggregate. The angle formed between the dashed line and the bottom of the experimental set up can be considered as the approximate underwater angle of repose. Figure 4j shows the underwater angle of repose $\theta_1$ against the downstream side and the underwater angle of repose $\theta_2$ against the upstream side.

Figure 4 also shows the process of the formation of the cone shaped deposition of aggregate. The aggregate settles into the water channel at a certain vertical velocity under gravitational force and accumulates at the bottom of the water channel. As the aggregate height is increased, the top of the deposited aggregate moves farther away from the perforated hole. At first, the cone shape is not obvious. As the aggregate continues to be poured into the tunnel, the cone shape is gradually more apparent. This cone shaped deposition of aggregate with a certain particle size in the water channel, which provides resistance to water flow in a section of the tunnel, acts as a bulkhead in the tunnel, blocking the water flow. The plugging section will resist water flow in the tunnel until the deposited height of the aggregate reaches the roof of the tunnel, so that, finally, the confined pipe flow can be converted to a seepage flow through the porous medium (aggregate). With the increase in poured aggregate, the height of the deposition also increases. The shape of the deposition becomes flatter in the horizontal direction. When the height of the deposited aggregate is close to the tunnel roof, the aggregate is washed away by water and accumulates in the downstream of the deposit. The top of the deposition is extended in the direction of the water flow.

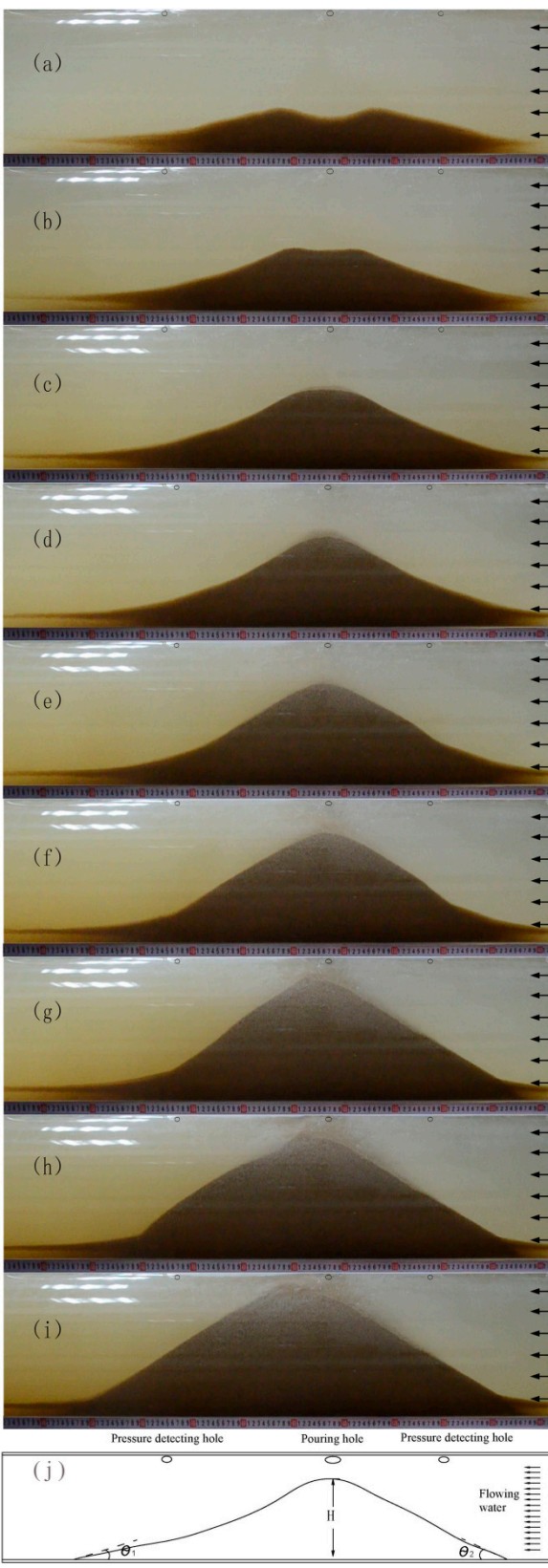

**Figure 4.** Series of diagrams on formation of deposit with cone shape. Poured aggregate: (**a**) 1500 g; (**b**) 3000 g; (**c**) 4500 g; (**d**) 6000 g; (**e**) 7500 g; (**f**) 9000 g; (**g**) 10,500 g (10.5 kg); (**h**) 12,000 g (12 kg); and (**i**) 13,500 g (13.5 kg). (**j**) Schematic diagram of angle of deposited aggregate with cone shape.

### 3.3. Factors that Influence Efficiency of Plugging

#### 3.3.1. Efficiency of Plugging

In this experiment, the cross section of the water channel was circular, and the accumulated mass of aggregate was cone shaped. When the deposited aggregate was at the maximum height, it could not completely be in close contact with the inside of the tunnel, thus resulting in a gap that appears between the deposition and the inside wall of the tunnel. The gap constitutes the residual water channel. A narrower residual channel means a better plugging. Figure 5 shows the shape of the residual water channel.

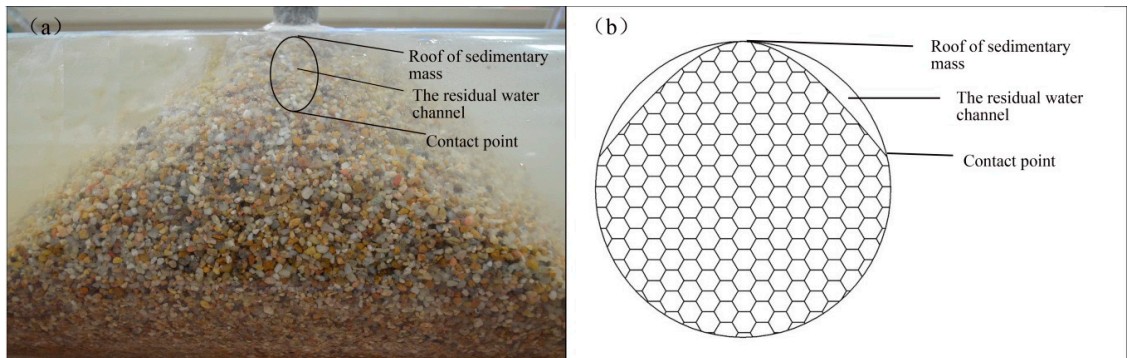

**Figure 5.** Shape of residual water channel: (**a**) photo; and (**b**) schematic diagram of cross section of deposition under pouring hole.

The efficiency of plugging (*PE*), as discussed in this paper, is defined as the percentage of the ratio of the maximum cross-section area of the deposited aggregate to the cross-section area of the tunnel. The formula is as follows:

$$PE\ (\%) = \frac{P_c}{P_0} \times 100 \tag{1}$$

where $P_c$ is the maximum cross-section area of the deposited aggregate and $P_0$ is the cross-section area of the tunnel.

In this study, the increased efficiency of plugging means a better seal.

#### 3.3.2. Main Effects

The range analysis of the efficiency of plugging and an analysis of the variance of the area of the residual water channel were applied to the experiments, and the following results were obtained.

Table 2 also lists the results on the efficiency of plugging and area of the residual water channel for each trial. The efficiency of plugging is inversely proportional to the area of the residual water channel. A smaller area of the residual channel means an increased efficiency of plugging and a better seal. The experimental results show that plugging with fine aggregate is better than with coarse aggregate.

Table 3 lists the data and range of each influencing factor. Figure 6 also clearly shows the efficiency of plugging with the main influencing factors which correspond to Table 3. It can be observed that the effectiveness of each factor is ranked in descending order as: the aggregate particle size (a range of 1.53, as shown in Table 3), initial velocity of the water flow (1.24), and water–solid mass ratio (0.65). Figure 6 shows that the optimal trial scheme is the value at the level of the maximum average *PE* reached, that is, *A1B1C3*. In the experiments, the optimal trial scheme was Trial 1 (*A1B1C1*), which was closest to the optimal trial scheme in Figure 6.

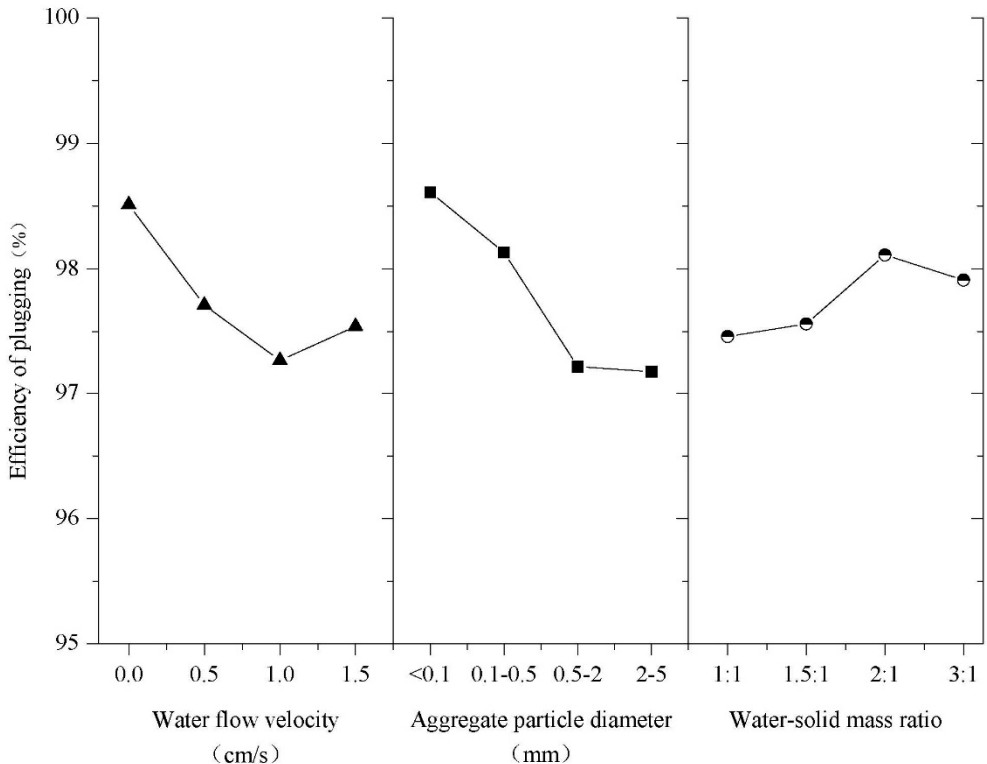

**Figure 6.** Response graphs of main effects in accordance with Table 3.

**Table 3.** Range analysis for main effects of plugging in tunnel with flow water.

| Four Levels | Average Efficiency of Plugging (%) for Factors | | |
| --- | --- | --- | --- |
| | Initial Velocity of Water Flow (cm/s) *A* | Particle Size of Aggregate (mm) *B* | Water–Solid Mass Ratio *C* |
| $PE_1$ | 98.51 | 98.61 | 97.46 |
| $PE_2$ | 97.71 | 98.13 | 97.56 |
| $PE_3$ | 97.27 | 97.22 | 98.11 |
| $PE_4$ | 97.54 | 97.18 | 97.91 |
| Range | 1.24 | 1.53 | 0.65 |

**Table 4.** Analysis of variance of cross-sectional area of residual water channel.

| Test Index | Variance Analysis Calculation | Value | | | |
| --- | --- | --- | --- | --- | --- |
| | | Initial Velocity of Water Flow *A* | Particle Size of Aggregate *B* | Water–Solid Mass Ratio *C* | Correction Error |
| The cross-section area of the residual water channel (mm²) | Deviation sum of squares | 274,692.40 | 520,620.19 | 89,490.71 | 31,438.2 |
| | Degree of freedom | 3 | 3 | 3 | 6 |
| | Mean square error | 91,564.13 | 173,540.06 | 29,830.24 | 5239.7 |
| | *F* ratio | 17.48 | 33.12 | 5.69 | – |

The analysis of variance of the area of the residual water channel is shown in Table 4. In the analysis of the index of the area of the residual water channel, it can be found that the analysis of variance of the *F* value of each factor is ranked in descending order as: the aggregate particle size (an *F* value of 33.12, as shown in Table 4), the initial velocity of the water flow (17.48), and water–solid mass ratio (5.69).

Among all of the influencing factors in the experiments, the influence of the particle size of aggregate *B* on the residual water channel is the most significant. According to the experimental results, finer particles result in a narrower residual water channel, and thus a better plugging.

### 3.3.3. Aggregate Particle Size and Initial Velocity of Water Flow

In the accumulation of aggregate particles, the interaction between the solid particles increases with increased concentration of aggregate particles in the tunnel. The surface of the deposition moves along the direction of the water flow and increases until reaching the top of the tunnel. It is considered that with increased initial velocity of water flow, the critical velocity of the aggregate can be easily reached, which results in a poor seal.

In comparison with coarse aggregate, the deposition of finer aggregate has a flatter surface and smaller angle of repose at the same flow velocity. The difference of the angle of slope between the upstream side and the downstream side is obvious. After pouring the aggregate, a larger particle will result in a larger repose angle, a wider residual water channel, and a poorer seal.

Figure 7 shows the curve of the water flow velocity for the four types of aggregate particles with increased amounts of aggregate. The change in the water flow velocity is exponential, and then approximately remains constant. When the maximum height of the deposited aggregate is reached, the residual channel remains the same and the water flow velocity also does not change.

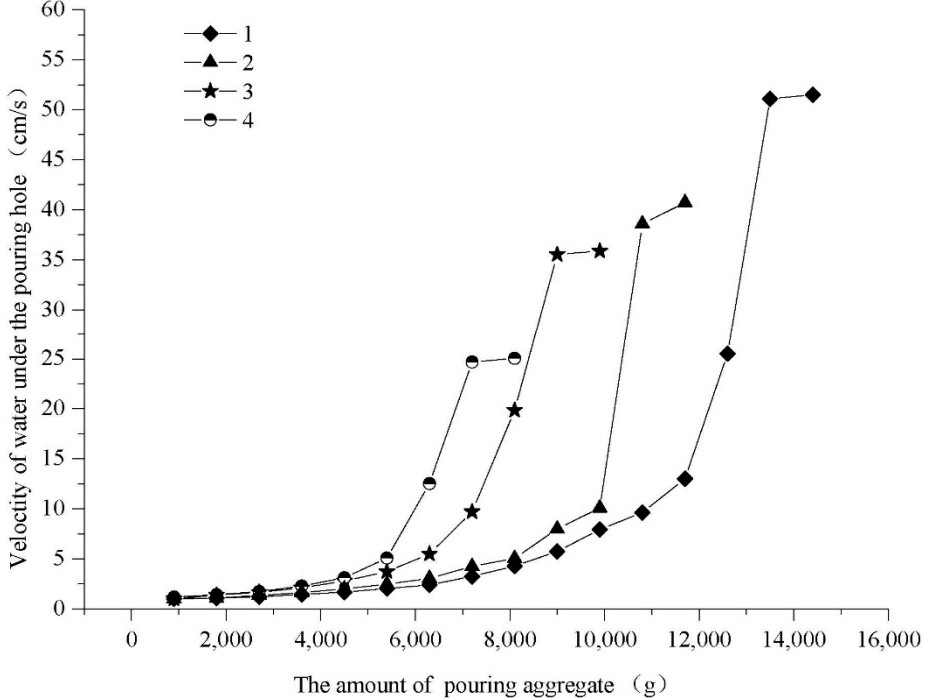

**Figure 7.** Velocity curves of aggregate with four different particle sizes under pouring hole. Aggregate particle size: 1, less than 0.1 mm; 2, 0.1–0.5 mm; 3, 0.5–2 mm; and 4, 2–5 mm.

Figure 7 shows that there is little difference in the initial velocity of water flow. With increased aggregate, the water flow velocity obviously changes. The flow velocity of the coarse aggregate with a particle size of 2–5 mm quickly changes, but the maximum velocity of the water flow, which is the maximum flow speed when the minimum residual channel is reached, under the pouring hole is less than that for the fine aggregate. It is considered that the influence of the aggregate particle size on the water flow velocity is relatively substantial in this experiment. According to the experiment by Duckworth and Argyros [21], when solid particles are heavier than water, this means that the coarse aggregate results in a greater critical velocity. The critical velocity is influenced by the density of solid materials, slurry concentration, and particle composition. The influence of particle composition on the critical velocity is fairly significant. The critical velocity of the coarse aggregate is greater than that of the fine aggregate, but the final velocity of the water flow when the cross section of accumulated aggregate tends stable is relatively low.

### 3.3.4. Water–Solid Mass Ratio

The water–solid mass ratio mainly affects the process of pouring of the aggregate. A small water–solid mass ratio is likely to plug the borehole, thus a larger water–solid mass ratio should be utilized when pouring aggregate. In this experiment, the water–solid mass ratio had little effect on the form of the final deposited shape regardless of the aggregate size. However, during the pouring, a larger water–solid mass ratio means that the aggregate is dispersed more quickly at the bottom of the tunnel. Compared with a low water–solid mass ratio, the speed at which the aggregate accumulates into a cone shaped mass is relatively low. Therefore, the water flow velocity and aggregate particle size should be applied for choosing an appropriate water–solid mass ratio in a tunnel with water flow to ensure that there will be no plugging of the pouring borehole and delaying the deposition.

## 4. Discussion

### 4.1. Criterion for Resistance to Flow

Critical velocity is an important parameter in slurry pipeline transport, which is related to the safety of pipeline transport itself. Slurry is composed of solid particles and water. If the conveying velocity is too low, the solid particles will settle, silt, and even plug the pipeline. The deposition of aggregate into a round type of pipe with water flow in this study is similar to slurry pipeline transport, and can therefore be examined along with the use of a critical velocity analysis. According to the experimental results, the critical flow velocity is concluded as the criterion for plugging in the section of the tunnel that resists water flow. When the deposition reaches the maximum height, the maximum flow velocity under the pouring hole is less than the critical velocity, which means that the section of the tunnel that resists water flow is plugged. The construction of this section of the tunnel provides the prerequisite for the subsequent grouting.

In accordance with Newton's laws, the moment that the particles just touch the bed and start to slide was studied in the experiments. The particles were analyzed as spherical in shape, and subjected to the conditions of the drag force $F_D$, lift force $F_L$, and the submerged gravitational force $W'$. Figure 8 shows a schematic diagram of these forces on the particles.

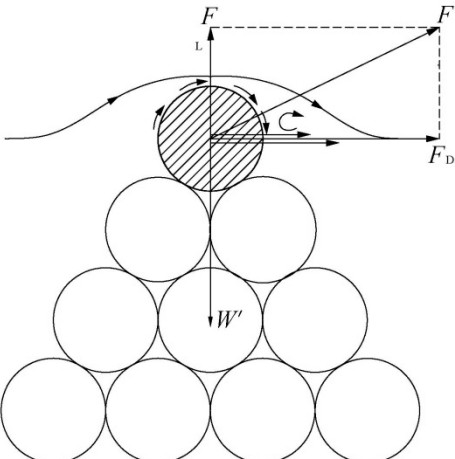

**Figure 8.** Schematic diagram of forces acting on particles.

The conditions for particle motion can generally be stated as follows:

$$F_D \geq f\left(W' - F_L\right) \tag{2}$$

Equation (2) can be further written as:

$$\rho u^2 (C_D - f\, C_L) \geq \frac{4}{3} f\, (\gamma_s - \gamma_w)\, d \tag{3}$$

where $\gamma_w$ is the unit weight of water, KN/m$^3$; $u$ is the velocity of water above particles, m/s; $C_D$ is the drag coefficient; $C_L$ is the lift force coefficient; $f$ is the friction coefficient between particles; $\gamma_s$ is the unit weight of solid particles, KN/m$^3$; and $d$ is the diameter of particle, m.

When the left and right sides of Equation (3) are equal, the particles are in the critical state where they will be transported without settling. Assuming that $u$ is equal to $u_c$ at this time, $u_c$ can be called the critical velocity, which is the minimum velocity of the particles in which they change from static to motion. Therefore, the critical velocity can be used as the criterion for the formation of the section of the tunnel that resists water flow. When the deposited aggregate reaches the maximum height, the section of the tunnel that resists water flow is plugged as long as the maximum flow velocity under the pouring hole is less than the critical velocity. The residual water channel is the dominant pathway of flow, and the velocity of water in this channel is much greater than that of the flow in the void, so that seepage velocity in the void can be neglected.

According to the analysis of the experiments, the critical velocity of fine aggregate is less than that of coarse aggregate. Aggregate particles settle and deposit due to friction loss in the pipe. When the pipe flow becomes infiltration flow, the section of the tunnel that resists water flow is basically plugged. The friction loss of the particles is mainly manifested in the friction between the particles and the pipe wall (or tunnel wall; friction coefficient is 0.36), particle motion loss (particle collision), etc.

The maximum flow velocity $u_{max}$ under the pouring hole for aggregate of four different particle sizes is shown in Figure 7. The maximum flow velocity is used in Equation (3) to determine the value of the left and right sides of Equation (3). If the condition in Equation (3) is satisfied, the section of the tunnel that resists water flow is not plugged. Table 5 lists the results of the criterion of the section of the tunnel that resists water flow that corresponds to Figure 7. Water flow is resisted when the deposited aggregate reaches the roof of the tunnel with a certain size, so that the confined pipe flow will be converted into seepage flow. However, due to the existence of the residual water channel, when the deposited aggregate reaches the roof of the tunnel, this also signifies that the tunnel is able to resist water flow, as evident from the tests. Table 5 shows that Equation (3) is feasible for this experiment. However, the parameters $C_D$, $C_L$, and $f$ are not easy to be defined in this equation. The well-known demi-McDonald nomogram for the limit deposition velocity of slurry based on the concept of a slitting bed of particles [35] or the modified formulas by Thomas [9] or Pinto et al. [10] for fully stratified slurry in horizontal pipes will be further investigated. Of course, the difference should be considered between the aggregate movement in plugging tunnel through a borehole and slurry flows in a horizontal pipe.

**Table 5.** Analysis of plug criterion of section of tunnel that resists water flow under dynamic state.

| Particle Size of Aggregate (mm) | $u_{max}$ (cm/s) | $\rho u^2 (C_D - f\, C_L)$ (N/m$^2$) | $\frac{4}{3} f\,(\gamma_s - \gamma_w) d$ (N/m$^2$) | Calculation Results of Criterion for Plugging | Successful or not for Plugging in the Experiments |
|---|---|---|---|---|---|
| 2–5 | 25 | 30.0 | 89.0 | Y | Y |
| 0.5–2 | 35 | 58.8 | 35.6 | N | N |
| 0.1–0.5 | 40 | 115.2 | 8.9 | N | N |
| <0.1 | 50 | 600.0 | 1.8 | N | N |

### 4.2. Optimal Spacing between Boreholes

The deposited aggregate settles in the tunnel with flow water due to friction loss, thus forming the section that resists flow and plugging the water source. If the water channel between adjacent pouring boreholes is filled with aggregate poured through the upstream boreholes, then the distance between the pouring boreholes comprises the optimal spacing. According to this experiment, the optimal spacing between the boreholes is affected by the anticipated volume, aggregate particle size, and water flow velocity. The concentration of fine aggregate is greater than that of coarse aggregate in the pipe, thus resulting in slower settlement than coarse particles. The aggregate particles are easily washed away by the water, as they have a low critical velocity. Thus, it is inferred that the optimal spacing between the boreholes for fine aggregate should be wider than that for coarse aggregate.

Figure 9 shows a schematic diagram of the optimum spacing between the boreholes. The deposited aggregate in the adjacent pouring boreholes can be categorized as the accumulated mass $V_2$ against the slope of the upstream, accumulated mass $V_1$ against the slope of the downstream, and the accumulated mass $V$ between the two adjacent pouring boreholes. Analysis of the optimum spacing between the boreholes $L$ can help in arranging the boreholes.

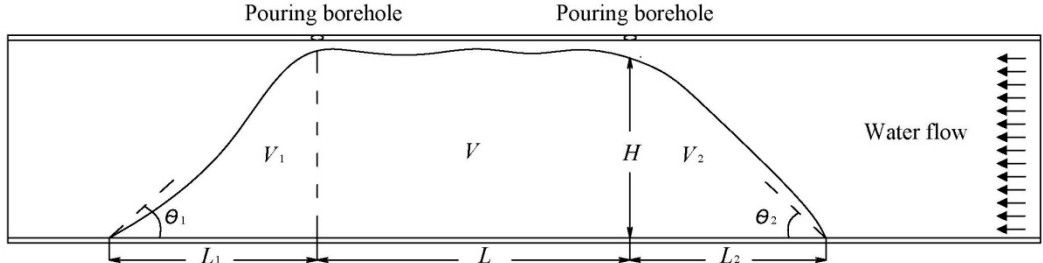

**Figure 9.** Schematic diagram of optimum spacing between boreholes.

### 4.3. Limitations

To clearly and intuitively observe grouting and plugging in a tunnel with water flow, a transparent acrylic pipe was used as a model of a tunnel in the experiment, but the inside of the tunnel is very smooth, and the roughness cannot be modeled to replicate the actual roughness of a real tunnel. The actual shape of the tunnel is also not just circular, and in the experiments, only a circular tunnel is examined. Other tunnel shapes should therefore be examined in future work. In addition, due to the experimental set-up and site restrictions, the flow velocity is less than that of the large water inrush found in a tunnel, and the water pressure is relatively low.

The results of the ranking of the different factors were merely a consequence of the ranges of variability of the corresponding operation parameters. The results are meaningful, because we determined the parameters with reference of practical engineering. Of course, the results may be different for different parameters. We will address this issue in further study.

Other influencing factors, such as the inclination angle of the tunnel, filling materials, and shape of the tunnel, will influence the plugging. The work in this study only examined the deposition of aggregate for plugging; however, the grouting as reinforcement in the next stage after deposition is also equally vital, and an important future work. Closely related to the examination of grouting is to consider more parameters in the experiments, such as water pressure before and after grouting, water flow velocity after grouting, and so on. These influencing factors will be investigated in our future work. The theoretical consideration from Equations (2) and (3) are neglecting the effect of seepage force, because the small hydraulic gradient results in a much smaller seepage velocity than the main water flow in the tunnel. In the further study, the seepage force exerted to the aggregate particles should be investigated. In addition, the drag, lift, and friction coefficients will be further determined in the future by considering the friction and inclination of tunnels.

## 5. Conclusions

In view of the obscurity problem in the pouring of aggregate into a tunnel, an experimental set up which can be used to model this process was designed in accordance with a similarity criterion of gravity. A series of experiments was carried in orthogonal arrays for pouring aggregate into a tunnel with flow water to examine the accumulation and diffusion of aggregate by using a model with a transparent tunnel replica. The three influencing factors tested in this experiment were the initial velocity of the water flow, aggregate particle size, and water–solid mass ratio. A range analysis and analysis of variance of the results show that the order of the influencing factors on the efficiency of plugging are the aggregate particle size, initial velocity of the water flow, and water–solid mass ratio.

In this experimental analysis, the settling of accumulated aggregate in a water channel was analyzed based on slurry pipeline transport. Aggregate settling is equivalent to the deposition of solid particles in slurry, and the particle settle under gravitational force, accumulating at the bottom of the tunnel due to the force of the water flow. Viewed at the side of model, the deposition is cone shaped. Compared to fine aggregate, the speed of settlement of coarse-grained aggregate is faster, and the deposited shape is relatively more cone-shaped. However, the cross-section area of the residual water channel is larger, with poor plugging.

The critical velocity of the water flow that can transport the aggregate without settling is regarded as the plugging criterion of the section of the tunnel that resists water flow. When the accumulated deposition reaches the maximum height, the maximum flow velocity under the pouring hole is less than the critical velocity, thus indicating that the section of the tunnel that resists water flow reaches the roof of the tunnel. The test results show that the critical velocity of fine aggregate is less than that of coarse aggregate, and the size of the section of the tunnel that resists water flow are larger. Meanwhile, the required minimum spacing for the formation of the section of the tunnel that resists water flow between boreholes is regarded as the optimum spacing, and an appropriate spacing can provide the theoretical basis for the layout of boreholes.

**Author Contributions:** G.Z. and S.H. contributed equally to this work. G.Z. and W.S. proposed the idea and designed the experiment. W.L. and S.H. carried out the experiments and prepared the draft. W.S. revised and modified the manuscript. All authors have read and agreed to the published version of the manuscript.

**Funding:** This study was supported by the Natural Science Foundation of China under Grant No. 4177283.

**Acknowledgments:** The authors would like to acknowledge the assistance from Yankun Liang and Shouliang Zhao, Guosheng Zheng, Shichong Yuan, Haiqing Liu, and Cong Xie of the CUMT for the help during testing and the editorial help from Jinchuan Zhang.

**Conflicts of Interest:** The authors declare no conflict of interest.

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
