# Peer review of "Experimental Investigation on Pouring Aggregate to Plug Horizontal Tunnel with Flow Water"

_water, doi:10.3390/w12061763_

Round 1

Reviewer 1 Report

My comments are reported in the attached document.

Reviewer 2 Report

This paper investigated the impact of particle size, flow velocity, and water-solid ratio on plugging of devised acrylic tunnel apparatus. The experiment is well designed and the manuscript is well written. The reviewer recommends to publish this paper in Water after addressing the following minor comments:

Title: I am not a native speaker, but to me, "water flow" sounds more natural than "flow water". I suggest to revise the title as follows:

"Experimental investigation on plugin horizontal tunnel with water flow by pouring aggregate"

Line 242: In this study, the increased effects of plugging points to a better seal. ==> what does this mean? hard for me to understand.

3.3.2. Main effects ==> please be more specific on title of this subsection. what does "main" here mean?

Round 2

Reviewer 1 Report

The authors satisfactorily addressed my comments, and the manuscript has significantly improved. I am pleased to recommend its acceptance in the Water journal in the present form. Just one note, in Ref. 19 the name of the second authors is Matoušek, not Natoušek.